# Towards Label-free Scene Understanding by Vision Foundation Models

**Runnan Chen**[1]    **Youquan Liu**[2]    **Lingdong Kong**[3]    **Nenglun Chen**[1]    **Xinge Zhu**[4]
**Yuexin Ma**[5*]    **Tongliang Liu**[6]    **Wenping Wang**[7*]
[1]The University of Hong Kong    [2]Hochschule Bremerhaven
[3]National University of Singapore    [4]The Chinese University of Hong Kong
[5]ShanghaiTech University    [6]The University of Sydney    [7]Texas A&M University

## Abstract

Vision foundation models such as Contrastive Vision-Language Pre-training (CLIP) and Segment Anything (SAM) have demonstrated impressive zero-shot performance on image classification and segmentation tasks. However, the incorporation of CLIP and SAM for label-free scene understanding has yet to be explored. In this paper, we investigate the potential of vision foundation models in enabling networks to comprehend 2D and 3D worlds without labelled data. The primary challenge lies in effectively supervising networks under extremely noisy pseudo labels, which are generated by CLIP and further exacerbated during the propagation from the 2D to the 3D domain. To tackle these challenges, we propose a novel Cross-modality Noisy Supervision (CNS) method that leverages the strengths of CLIP and SAM to supervise 2D and 3D networks simultaneously. In particular, we introduce a prediction consistency regularization to co-train 2D and 3D networks, then further impose the networks' latent space consistency using the SAM's robust feature representation. Experiments conducted on diverse indoor and outdoor datasets demonstrate the superior performance of our method in understanding 2D and 3D open environments. Our 2D and 3D network achieves label-free semantic segmentation with 28.4% and 33.5% mIoU on ScanNet, improving 4.7% and 7.9%, respectively. For nuImages and nuScenes datasets, the performance is 22.1% and 26.8% with improvements of 3.5% and 6.0%, respectively. Code is available.[2]

## 1   Introduction

Scene understanding aims to recognize the semantic information of objects within their contextual environment, which is a fundamental task for autonomous driving, robot navigation, digital city, etc. Existing methods have achieved remarkable advancements in 2D and 3D scene understanding [1–10]. However, they heavily rely on extensive annotation efforts and often struggle to identify novel object categories that were not present in the training data. These limitations hinder their practical applicability in real-world scenarios where acquiring high-quality labelled data can be expensive and novel objects may appear [11–17]. Consequently, label-free scene understanding, which aims to perform semantic segmentation in real-world environments without requiring labelled data, emerges as a highly valuable yet relatively unexplored research topic.

Vision foundation models, *e.g.*, Contrastive Vision-Language Pre-training (CLIP) [18] and Segment Anything (SAM) [19], have garnered significant attention due to their remarkable performance in addressing open-world vision tasks. CLIP, trained on a large-scale collection of freely available image-text pairs from websites, exhibits promising capabilities in open-vocabulary image classification. On

---

*Corresponding authors.
[2]https://github.com/runnanchen/Label-Free-Scene-Understanding.

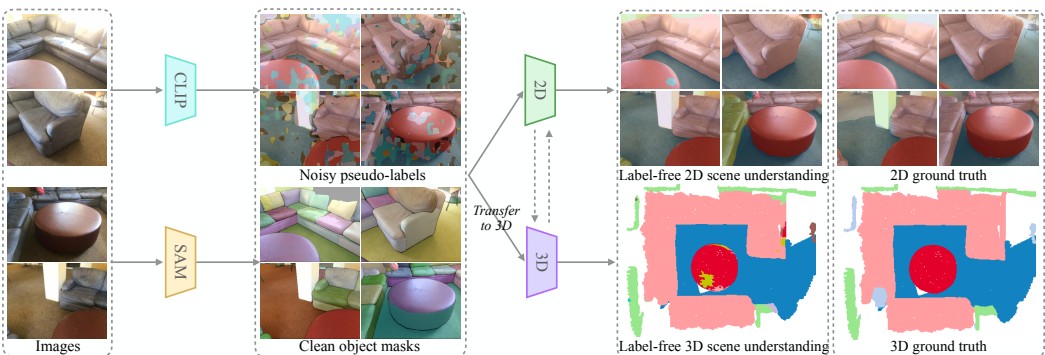

Figure 1: We study how vision foundation models enable networks to comprehend 2D and 3D environments without relying on labelled data. To accomplish this, we introduce a novel framework called Cross-modality Noisy Supervision (CNS). By effectively harnessing the strengths of CLIP and SAM, our approach simultaneously trains 2D and 3D networks, yielding remarkable performance.

the other hand, SAM learns from an extensive dataset comprising 1 billion masks across 11 million images, achieving impressive zero-shot image segmentation performance. However, initially designed for image classification, CLIP falls short in segmentation performance. Conversely, SAM excels in zero-shot image segmentation but lacks object semantics (Fig. 1). Additionally, both models are trained exclusively on 2D images without exposure to any 3D modal data. Given these considerations, a natural question arises: Can the combination of CLIP and SAM imbue both 2D and 3D networks with the ability to achieve label-free scene understanding in real-world open environments?

Despite recent efforts [20] leverage CLIP for image-based semantic segmentation, the pseudo labels generated by CLIP for individual pixels often exhibit significant noise, resulting in unsatisfactory performance. The up-to-date work [17] has extended this method to encompass 3D label-free scene understanding by transferring 2D knowledge to 3D space via projection. However, the pseudo labels assigned to 3D points are considerably noisier due to calibration errors, significantly limiting the accuracy of the networks. The primary challenge of utilizing vision foundation models for scene understanding is effectively supervising the networks using exceptionally noisy pseudo labels.

Inspired by SAM's impressive zero-shot segmentation capabilities, we propose a novel Cross-modality Noisy Supervision (CNS) framework incorporating CLIP and SAM to train 2D and 3D networks simultaneously. Specifically, we employ CLIP to densely pseudo-label 2D image pixels and transfer these labels to 3D points using the calibration matrix. Since pseudo-labels are extremely noisy, leading to unsatisfactory performance, we refine pseudo-labels using SAM's masks, producing more reliable pseudo-labels for supervision. To further mitigate error propagation and prevent the networks from overfitting the noisy labels, we consistently regularize the network predictions, *i.e.*, co-training the 2D and 3D networks using the randomly switched pseudo labels, where the labels are from the prediction of 2D, 3D, and CLIP networks. Moreover, considering that individual objects are well distinguished in SAM feature space, we use the robust SAM feature to consistently regularize the latent space of 2D and 3D networks, which aids networks in producing semantic predictions with more precise boundaries and further reduces label noise. Note that the SAM feature space is frozen during training, thus severed as the anchor metric space for aligning the 2D and 3D features.

To verify the label-free scene understanding capability of our method in 2D and 3D real open worlds, we conduct experiments on indoor and outdoor datasets, *i.e.*, ScanNet and nuScenes, where 2D and 3D data are simultaneously collected. Extensive results show that our method significantly outperforms state-of-the-art methods in understanding 2D and 3D scenes without training on any labelled data. Our 2D and 3D network achieves label-free semantic segmentation with 28.4% and 33.5% mIoU on ScanNet, improving 4.7% and 7.9%, respectively. For nuImages and nuScenes datasets, the performance is 22.1% and 26.8% with improvements of 3.5% and 6.0%, respectively. Quantitative and qualitative ablation studies also demonstrate the effectiveness of each module.

The key contributions of our work are summarized as follows.

- In this work, we study how vision foundation models enable networks to comprehend 2D and 3D environments without relying on labelled data.

- We propose a novel Cross-modality Noisy Supervision framework to effectively and synchronously train 2D and 3D networks with severe label noise, including prediction consistency and latent space consistency regularization schemes.

- Experiments conducted on indoor and outdoor datasets show that our method significantly outperforms state-of-the-art methods on 2D and 3D semantic segmentation tasks.

## 2 Related Work

**Scene Understanding.** Scene understanding is a fundamental task for computer vision and plays a critical role for robotics, autonomous driving, smart city, etc. Significant advancements have been made by current supervised methods for 2D and 3D scene understanding [1–10, 21–34]. However, these methods heavily depend on extensive annotation efforts, which pose significant challenges when encountering novel object categories that were not included in the training data. In order to overcome these limitations, researchers have proposed self-supervised and semi-supervised methods [13, 14, 35–54] to train networks in a more data-efficient manner. Nevertheless, these approaches struggle to handle novel objects that are not present in training data and they often exhibit subpar performance when faced with significant domain gaps. Alternatively, some methods [11, 12, 55–64] focus on open-world scene understanding that identifies novel object categories absents in the training data. However, these methods still require expensive annotation efforts on the close-set category data. Other methods [56, 17, 65–67] try to alleviate 3D annotation efforts by distilling knowledge from 2D networks. However, they either need 2D labelled data to train networks or struggle to achieve satisfactory results in the context of label-free 3D scene understanding.

Recently, vision foundation models such as Contrastive Vision-Language Pre-training (CLIP) [18] and Segment Anything (SAM) [19] have garnered significant attention due to their remarkable performance in open-world vision tasks. MaskCLIP [20] extends the capabilities of CLIP to image semantic segmentation, achieving promising results in 2D scene understanding without labelled data. Furthermore, CLIP2Scene [17] expands upon MaskCLIP to enable 3D scene understanding through the use of a 2D-3D calibration matrix. In this study, our focus lies in label-free understanding using vision foundation models. Our ultimate goal is to perform semantic segmentation in real-world environments without relying on labelled data. We hope that our work will inspire and motivate further research in this area.

**Label-noise Representation Learning.** Label-Noise Representation Learning (LNRL) aims to train neural networks robustly in the presence of noisy labels. Numerous methods have been developed to tackle this challenge. Some methods [68–74] focus on designing accurate estimators of the noise transition matrix, which establish a connection between the clean class posterior and the noisy class posterior. Others [75–81] devise noise-tolerant objective functions through explicit or implicit regularization, enabling the learning of robust classifiers on noisy data. Furthermore, Some approaches [82–86] delve into the dynamic optimization process of LNRL based on the memorization effects, *i.e.*, deep models tend to initially fit easy (clean) patterns and gradually overfit complex (noisy) patterns. However, the above methods are insufficient when clean labels are unavailable, which makes them inadequate for addressing our problem with entirely noisy labels. Furthermore, these methods are primarily designed for a single modality, and the exploration of multiple modality scenarios is still an unexplored topic. In this paper, we study the LNRL across different modalities without any clean labels.

## 3 Methodology

The vision foundation models have demonstrated remarkable performance in open-world scenarios. However, while CLIP excels in zero-shot image classification, it tends to produce noisy masks in segmentation tasks. Conversely, SAM generates impressive object masks in zero-shot segmentation but lacks object semantics. Therefore, our research aims to mitigate the impact of noisy labels by leveraging the strengths of various modalities, such as images, point clouds, CLIP, and SAM. In what follows, we revisit the vision foundation models applied in open-vocabulary classification and semantic segmentation, then present the Cross-modality Noisy Supervision (CNS) framework in detail.

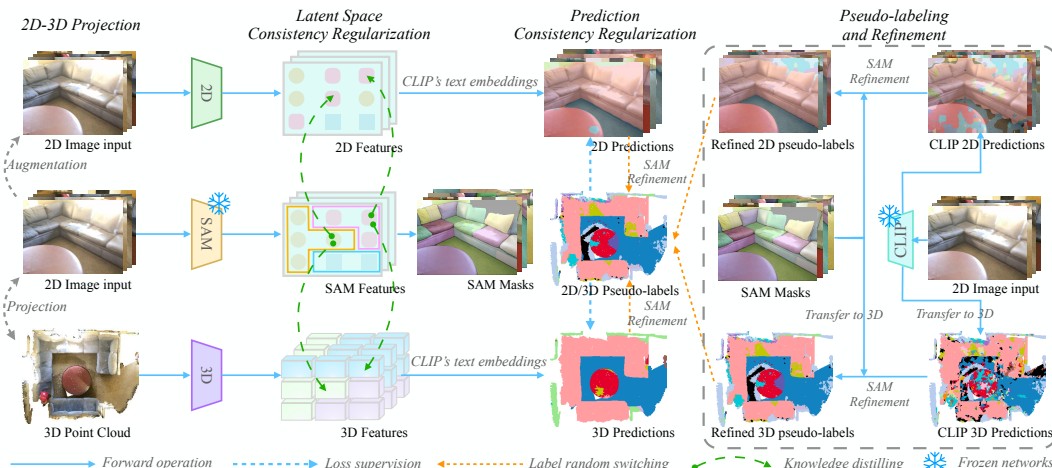

Figure 2: Illustration of the overall framework. Specifically, we employ CLIP to densely pseudo-label 2D image pixels and transfer these labels to 3D points using the calibration matrix. Acknowledging that network predictions and pseudo-labels are extremely noisy, we refine pseudo-labels using SAM's masks for more reliable supervision. To mitigate error propagation and prevent the networks from overfitting the noisy labels, we randomly switch the different refined pseudo-labels to co-training the 2D and 3D networks. Besides, we consistently regularize the network's latent space using SAM's robust feature, which benefits the networks to produce clearer predictions.

## 3.1 Revisiting Vision Foundation Models

Vision foundation model refers to powerful models pre-trained on large-scale multi-modal data and can be applied to a wide range of downstream tasks [87]. CLIP and SAM are well recognized due to their impressive open-world image classification and segmentation performance.

CLIP combines an image encoder (ResNet [88] or ViT [89]) and a text encoder (Transformer [90]), which independently map image and text representations to a shared embedding space. CLIP trains both encoders using a contrastive loss on a dataset of 400 million image-text pairs sourced from the Internet that contain diverse image classes and textual concepts. This training approach enables CLIP to achieve impressive performance in open-vocabulary recognition. CLIP utilizes a two-step process for zero-shot image classification. Initially, it obtains image embeddings and generates text embeddings by incorporating the class name into a predefined template. Then, it compares the similarity of image and text embeddings to determine the class. SAM serves as the fundamental model for image segmentation. It undergoes training using over 1 billion masks on a dataset of 11 million licensed images, enabling it to adapt seamlessly to new image distributions and tasks without prior exposure. Extensive evaluations have demonstrated that SAM's zero-shot performance is remarkable, frequently rivaling or surpassing the results of previous fully supervised approaches. Note that SAM works impressively on zero-shot image segmentation but lacks object semantics.

MaskCLIP [20] extends the CLIP's capabilities to image semantic segmentation. Specifically, MaskCLIP modifies the attention pooling layer of CLIP's image encoder to enable pixel-level mask prediction instead of global image-level prediction. CLIP2Scene [17] further expands MaskCLIP into 3D scene understanding by transferring the pixel-level pseudo-labels from the 2D image to the 3D point cloud using a 2D-3D calibration matrix. However, their performance is unsatisfactory due to the significant noise in the pseudo labels. To this end, we leverage the strengths of vision foundation models to supervise networks under extremely noisy pseudo labels.

## 3.2 Cross-modality Noisy Supervision

As illustrated in Fig. 2, we introduce a novel Cross-modality Noisy Supervision (CNS) method that combines CLIP and SAM foundation models to concurrently train 2D and 3D networks for label-free scene understanding. Our approach involves pseudo-labelling the 2D images and 3D points using CLIP. To address the noise in pseudo-labelling, we employ SAM's masks to refine the pseudo-labels. To further mitigate error propagation and prevent overfitting to noisy labels, we

propose a prediction consistency regularization mechanism. This mechanism randomly switches refined pseudo-labels from different modalities to co-train the 2D and 3D networks. Additionally, we consistently regularize the network's latent space using SAM's robust feature, which aids in generating more accurate predictions. In the following sections, we provide a comprehensive overview of our method, along with detailed explanations and insights.

**Pixel-point Projection.** Since vision foundation models are pre-trained on 2D images, we transfer knowledge from 2D images to 3D point clouds via pixel-point correspondences. Note that the dataset already provides the camera poses, so we obtain the pixel-point correspondence $\{c_i, s_i, x_i, p_i\}_{i=1}^N$ via simple projection, where $c_i$, $s_i$ and $x_i$ indicates the $i$-th paired image pixel feature by CLIP, SAM, and 2D network, respectively. $p_i$ is the $i$-th paired point feature by the 3D network $\mathcal{F}_{3D}$ and $N$ is the number of pairs.

**Pseudo-labeling by CLIP.** We follow the pioneer methods, MaskCLIP and CLIP2Scene, that adopt CLIP to dense pseudo-label the 2D image pixels and transfer these labels to 3D points. Specifically, given the calibrated pixel-point features $\{c_i, s_i, x_i, p_i\}_{i=1}^N$, we pseudo-label the pixel $c_i$ as follows:

$$P_{c_i} = \arg\max_l \psi(c_i^T, t_l), \tag{1}$$

where $c_i$ is the $i$-th pixel feature by CLIP's image encoder. $t_l$ is the CLIP's text embedding of the $l$-th class name into a pre-defined template. $\psi$ is the dot product operation.

Since pixel-point features $\{c_i, s_i, x_i, p_i\}_{i=1}^N$ are spatial aligned, we can transfer the pseudo-label $\{P_{c_i}\}_{i=1}^N$ to all image pixels and point coulds, *i.e.*,

$$P_{s_i} = P_{x_i} = P_{p_i} = P_{c_i}, \tag{2}$$

where $\{P_{s_i}\}_{i=1}^N$, $\{P_{x_i}\}_{i=1}^N$ and $\{P_{p_i}\}_{i=1}^N$ are the transferred pseudo-labels from CLIP to SAM, 2D and 3D networks, respectively.

**Label Refinement by SAM.** Since CLIP is pre-trained on 2D images and text for zero-shot classification, it brings extreme noise to pixel-wise prediction. Nevertheless, another vision foundation model, Segment Anything (SAM), can provide impressive object masks in open-world environments, even rivaling or surpassing the previous fully supervised approaches. To this end, we consider refining the CLIP's pseudo-labels by SAM. Firstly, we follow the official implementation of SAM to generate all masks from the images. The mask ID of pixels are presented as $\{M_{s_i}\}_{i=1}^N$, where $M_{s_i} \in \{1, 2, ..., T\}$ and $T$ indicates the number of masks. Once again, we could transfer the masks to all image pixels and point clouds based on the pixel-point correspondence $\{c_i, s_i, x_i, p_i\}_{i=1}^N$, *i.e.*,

$$M_{c_i} = M_{x_i} = M_{p_i} = M_{s_i}, \tag{3}$$

where $\{M_{c_i}\}_{i=1}^N$, $\{M_{x_i}\}_{i=1}^N$ and $\{M_{p_i}\}_{i=1}^N$ are the transferred masks from SAM to CLIP , 2D and 3D predictions, respectively.

We then adjust the pseudo-labels to be identical if they are within the same mask. For simplicity, we take the max voting strategy, *i.e.*,

$$P_{c_i}^* = \arg\max_l \sum_{c_j, M_{c_i}=M_{c_j}} \mathbb{1}\{P_{c_j} = l\}, \tag{4}$$

where $\mathbb{1}\{\cdot\}$ is the indicator function. $P_{c_i}^*$ is the SAM refined pseudo-label from $P_{c_i}$. To this end, all pixels and point pseudo-labels within the same mask will be corrected by the semantic label most pixels belonging to, *i.e.*, $\{P_{c_i}^*\}_{i=1}^N$, $\{P_{x_i}^*\}_{i=1}^N$ and $\{P_{p_i}^*\}_{i=1}^N$.

**Prediction Consistency Regularization.** After obtaining the refined pseudo labels of all image pixels and points $\{P_{c_i}^*\}_{i=1}^N$, $\{P_{x_i}^*\}_{i=1}^N$ and $\{P_{p_i}^*\}_{i=1}^N$, we consider how to train both 2D and 3D networks. The training process consists of two stages. In the first stage, we simultaneously supervise the 2D and 3D networks by $\{P_{x_i}^*\}_{i=1}^N$ and $\{P_{p_i}^*\}_{i=1}^N$ with Cross-entropy loss:

$$\mathcal{L}_{s1} = \sum_i CE(\theta_s(x_i), P_{x_i}^*) + \sum_i CE(\varphi_s(p_i), P_{p_i}^*), CE(x, y) = -\log \frac{\exp(\psi(x, t_y))}{\sum_l \exp(\psi(x, t_l))}, \tag{5}$$

where $\theta_s$ and $\varphi_s$ are linear mapping layers of 2D $\{x_i\}_{i=1}^N$ and 3D $\{p_i\}_{i=1}^N$ features, respectively. $t_l$ is the CLIP's text embedding of the $l$-th class name. Note that CLIP and 2D network inputs are the

same images with different data augmentation. Thus we can utilize all CLIP's pseudo labels to train the 2D network.

In the second stage, we additionally include the self-training and cross-training process to reduce the error propagation flow of noisy prediction. Specifically, we first obtain the self-predicted pseudo-labels of 2D network $\hat{P}_{x_i}^*$ and 3D network $\hat{P}_{p_i}^*$, respectively. Note that $\hat{P}_{x_i} = \arg\max_l \psi(\theta_s(x_i), t_l)$, $\hat{P}_{p_i} = \arg\max_l \psi(\varphi_s(p_i), t_l)$ and then they are refined to be $\hat{P}_{x_i}^*$ and $\hat{P}_{p_i}^*$ by SAM (Formula 4). Next, we randomly feed the 2D, 3D, and refined CLIP pseudo-labels to train the 2D and 3D networks. The objective function is as follows:

$$\mathcal{L}_{s1} = \sum_i CE(\theta_s(x_i), t_{R_i^*}) + \sum_i CE(\varphi_s(p_i), t_{R_i^*}), \tag{6}$$

where the pseudo-label $R_i^* \in \{P_{x_i}^*, P_{p_i}^*, \hat{P}_{x_i}^*, \hat{P}_{p_i}^*\}$ is randomly assigned to 2D and 3D networks with equal possibility during training.

Note that two training stages are seamless, *i.e.*, we train 2D and 3D networks in the first stage with a few epochs to warm up the network and then seamlessly switch to the second training stage. In this way, we reduce the error propagation flow of noisy prediction and prevent the networks from overfitting the noisy labels.

**Latent Space Consistency Regularization**    Inspired by the remarkable performance of zero-shot segmentation achieved by SAM, where objects are usually distinguished with clear boundaries, we consider how SAM endows the 2D and 3D networks with the robust segment anything capability to further resist the prediction noise. The intuitive solution is to transfer the knowledge from the SAM feature space to the image and point cloud feature spaces learned by target 2D and 3D networks.

Previous methods [17, 14, 13] employ InfoNCE loss for cross-modal knowledge transfer. These methods initially establish positive pixel-point pairs and negative pairs. They subsequently employ the InfoNCE loss to encourage the positive pairs to be closer and push away the negative pairs in the embedding space. However, Because precise object semantics are missing, these methods face a common optimization-conflict challenge. For instance, if two pixels have identical semantics but are wrongly assigned to different clusters, the InfoNCE loss tends to separate them, which is unreasonable and adversely affects downstream task performance [17, 14]. To this end, we choose to directly pull in the point feature with its corresponding pixel in the SAM feature space. The objective function is:

$$\mathcal{L}_f = \sum_{i=1}^N (1 - COS(\theta_f(x_i), \frac{\phi_f(s_i)}{\|\phi_f(s_i)\|_2})) + (1 - COS(\varphi_f(p_i), \frac{\phi_f(s_i)}{\|\phi_f(s_i)\|_2})), \tag{7}$$

where $\theta_f(\cdot)$, $\varphi_f(\cdot)$, and $\phi_f(\cdot)$ indicate the linear mapping layers of 2D $\{x_i\}_{i=1}^N$, 3D $\{p_i\}_{i=1}^N$ and SAM feature $\{s_i\}_{i=1}^N$, with the same output feature dimensions $K_f$. $COS(\cdot)$ is the cosine similarity. Note that the SAM feature is frozen during training.

Essentially, SAM has been trained on millions of images using large-scale masks, resulting in a highly robust feature representation contributing to its exceptional performance in zero-shot image segmentation. This robust feature representation can be used as a foundation for aligning the 2D and 3D features so that 2D and 3D paired features can be constrained into a single, unified SAM feature space to alleviate the problem of noisy predictions.

## 4 Experiments

To evaluate the superior performance and generalization capability of our method for scene understanding, we conduct experiments on both indoor and outdoor public datasets, namely ScanNet [91], nuScenes [92] and nuImages [93], and compare with current state-of-the-art approaches [20, 56, 17] on both 2D and 3D semantic segmentation tasks. Critical operations and losses of our method are also verified effectiveness by extensive ablation studies. In the following, we introduce datasets, implementation, comparison results, and ablation studies in detail.

**Datasets.**    ScanNet [91] consists of 1,603 indoor scans, collected by RGB-D camera, with 20 classes, where 1,201 scans are allocated for training, 312 scans for validation, and 100 scans for testing. Additionally, we utilize 25,000 key frame images to train the 2D network. The nuScenes [92] dataset,

Table 1: Comparison (mIoU) with current state-of-the-art label-free methods for semantic segmentation tasks on the ScanNet [91], nuImages [93] and nuScenes [94] dataset.

| Methods | Publication | ScanNet 2D | ScanNet 3D | nuImages 2D | nuScenes 3D |
|---|---|---|---|---|---|
| MaskCLIP [20] | ECCV 2022 | 17.3 | 14.2 | 14.1 | 12.8 |
| MaskCLIP+ [20] | ECCV 2022 | 20.3 | 21.6 | 17.3 | 15.3 |
| OpenScene [56] | CVPR 2023 | 14.2 | 16.8 | 12.4 | 14.6 |
| CLIP2Scene [17] | CVPR 2023 | 23.7 | 25.6 | 18.6 | 20.8 |
| Ours | – | **28**.4 | **33**.5 | **22**.1 | **26**.8 |

Table 2: Ablation study on the ScanNet [94] *val* set for label-free 2D and 3D semantic segmentation.

| Ablation Target | Setting | 2D mIoU (%) | 3D mIoU (%) |
|---|---|---|---|
| Cross-modality Noisy Supervision | Baseline | 17.3 | 14.2 |
| | w/o CNS | 24.5 $_{(+7.20)}$ | 19.4 $_{(+5.20)}$ |
| | Base+CLIP | 20.4 $_{(+3.10)}$ | 23.8 $_{(+9.60)}$ |
| | Base+CLIP+LSC | 22.5 $_{(+5.20)}$ | 28.7 $_{(+14.5)}$ |
| Label Refinement | w/o Refine | 23.7 $_{(+6.40)}$ | 30.6 $_{(+16.4)}$ |
| | Post-Refine | 28.1 $_{(+10.8)}$ | 33.2 $_{(+19.0)}$ |
| Prediction Consistency Regularization | w/o CT | 27.7 $_{(+10.1)}$ | 31.0 $_{(+16.8)}$ |
| | w/o SCT | 24.7 $_{(+7.40)}$ | 30.6 $_{(+16.4)}$ |
| | w/o CLIP | 4.7 $_{(-12.6)}$ | 5.1 $_{(-9.1)}$ |
| | w 2D | 6.9 $_{(-10.4)}$ | 8.8 $_{(-5.4)}$ |
| | w 3D | 4.8 $_{(-12.5)}$ | 6.4 $_{(-7.8)}$ |
| Latent Space Consistency Regularization | CLIP | 27.3 $_{(+10.0)}$ | 32.4 $_{(+18.2)}$ |
| | SAM+CLIP | 27.8 $_{(+10.5)}$ | 32.5 $_{(+18.3)}$ |
| | None | 27.1 $_{(+9.80)}$ | 32.0 $_{(+17.8)}$ |
| Image Numbers | 8 | 27.9 $_{(+10.6)}$ | 33.0 $_{(+18.8)}$ |
| | 16 | 28.4 $_{(+11.1)}$ | 33.5 $_{(+19.3)}$ |
| | 32 | 28.3 $_{(+11.0)}$ | 33.3 $_{(+19.1)}$ |
| Full Configuration | Ours | **28.4** $_{(+11.1)}$ | **33.5** $_{(+19.3)}$ |

collected in traffic scenarios by LiDAR and RGB camera, comprises 700 scenes for training, 150 for validation, and 150 for testing, focusing on LiDAR semantic segmentation with 16 classes. To be more specific, we leverage a total of 24,109 sweeps of LiDAR scans for training and 4,021 sweeps for validation. Each sweep is accompanied by six camera images, providing a comprehensive 360-degree view. Note that the dataset does not provide image labels. Thus the quantitative evaluation of the paired images is not presented. The nuImages [93] dataset provides 93,000 2D annotated images sourced from a significantly larger dataset. This includes 67,279 images for training, 16,445 for validation, and 9,752 for testing. The dataset consists of 11 shared classes with nuScenes.

**Implementation Details.** We utilize MinkowskiNet34 [95] as the 3D backbone and DeeplabV3 [96] as the 2D backbone in our approach, where DeeplabV3 model is pre-trained on the ImageNet dataset. Based on the MaskCLIP framework, we modify the attention pooling layer of CLIP's image encoder to extract dense pixel-text correspondences. Our framework is developed using PyTorch and trained on two NVIDIA Tesla A100 GPUs. During training, both CLIP and SAM are kept frozen. For prediction consistency regularization, we transition to stage two after ten epochs of stage one. To enhance the robustness of our model, we apply various data augmentations, such as random rotation along the z-axis and random flip for point clouds, as well as random horizontal flip, random crop, and random resize for images. For the ScanNet dataset, the training process takes approximately 10 hours for 30 epochs, with the image number set to 16. In the case of the nuScenes dataset, the training time is 40 hours for 20 epochs, with the image number set to 6.

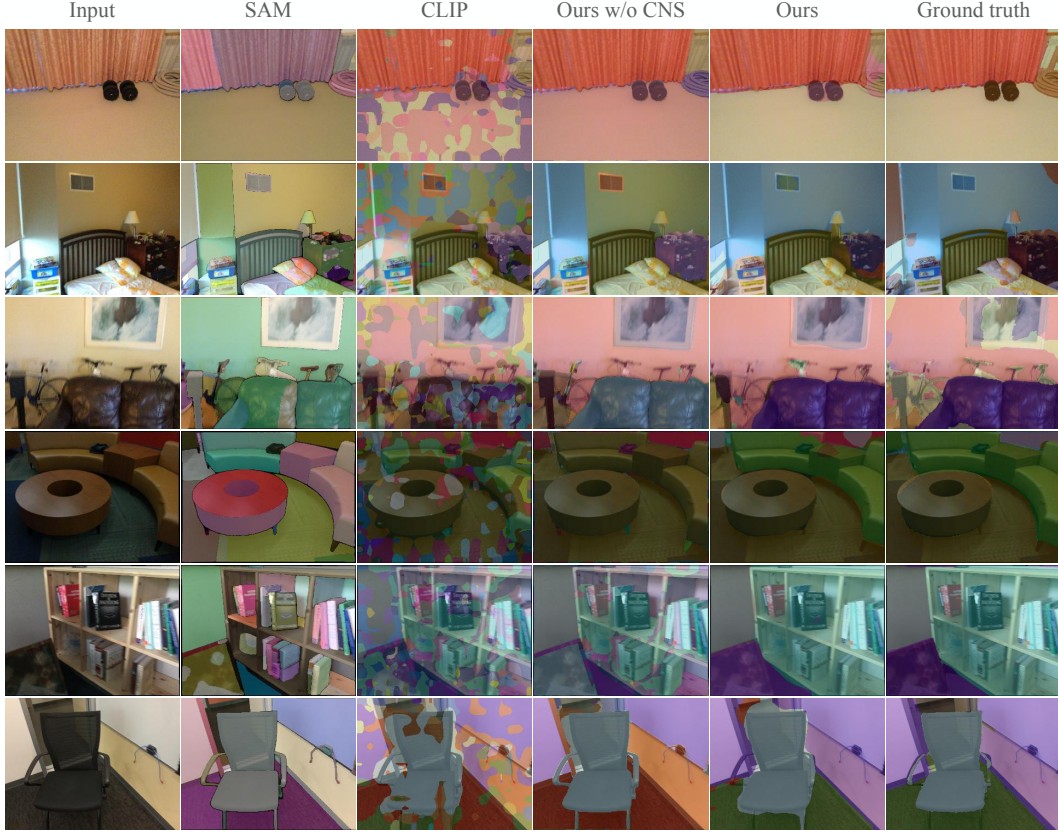

| Input | SAM | CLIP | Ours w/o CNS | Ours | Ground truth |

Figure 3: Qualitative results of label-free semantic segmentation on the ScanNet 25k images dataset. From the left to the right column are the input, prediction by SAM, CLIP, ours w/o CNS, our full method, and ground truth, respectively.

## 4.1 Comparison Results

Our method was compared with several state-of-the-art methods, namely MaskCLIP, MaskCLIP+, OpenScene, and CLIP2Scene, on the ScanNet and nuScenes datasets. Note that their codes are publicly available. For a fair comparison, we share all experiment settings with the above methods, including data augmentations, training epochs, learning rate, and network backbones. The results, as depicted in Table 1, demonstrate that our method surpasses the performance of other approaches by a significant margin. In terms of label-free semantic segmentation, our 2D and 3D networks achieve remarkable results, achieving 28.4% and 33.5% mIoU on the ScanNet dataset, corresponding to improvements of 4.7% and 7.9%, respectively. Furthermore, on the nuScenes dataset, our method achieves a mIoU of 26.8% for 3D semantic segmentation, exhibiting a 6% improvement. Note that the nuScenes dataset does not provide image labels. Thus the quantitative evaluation of 2D semantic segmentation is not presented.

## 4.2 Ablation Study

We conduct a series of ablation experiments to evaluate and demonstrate the effectiveness of different modules within our framework, as shown in Table. 2. In this section, we present the ablation configuration and provide an in-depth analysis of our experimental results.

**Effect of Cross-modality Noisy Supervision.** In 2D scene understanding, the **Baseline** is CLIP. For 3D scene understanding, we directly project the 2D prediction to the 3D point cloud. We also conducted **w/o CNS** that the Cross-modality Noisy Supervision was not applied, *i.e.*, build upon Baseline, the 2D label is refined by SAM, and projected to 3D points. Besides, we impose the baseline method self-training with the CLIP pseudo-labels, termed **base+CLIP**. Building upon **base+CLIP**, we also impose Latent Space Consistency Regularization, termed **base+CLIP+ LSCR**. Experiments

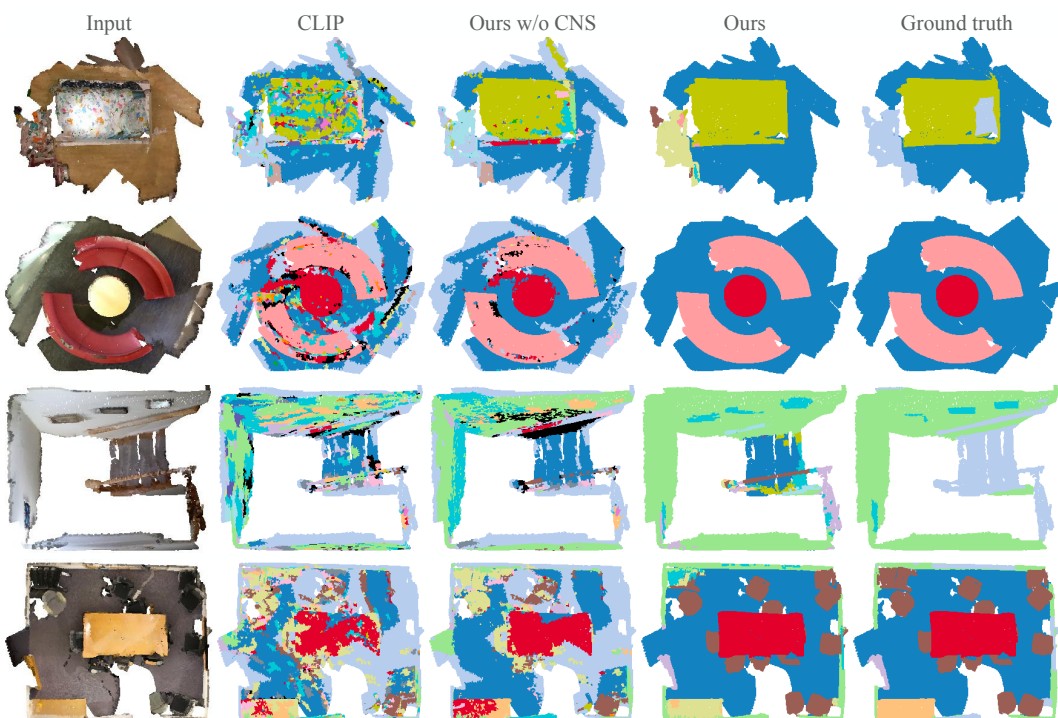

Figure 4: Qualitative results of label-free point cloud-based 3D semantic segmentation on the ScanNet dataset. From the left to the right are the input, prediction by CLIP, ours w/o CNS, our full method, and ground truth, respectively.

show that SAM refinement promotes pseudo-label quality. Moreover, our Cross-modality Noisy Supervision can effectively supervise networks under extremely noisy pseudo labels. We also evaluate CNS qualitatively using the ScanNet dataset, as illustrated in Fig. 3 and 4. Our comprehensive approach demonstrates remarkable performance in label-free scene understanding, encompassing both 2D and 3D scenarios. We observed that our results are comparable to human annotations in numerous instances. Interestingly, the 2D and 3D networks exhibit the remarkable ability to "segment anything," which can be attributed to the knowledge transferred from SAM.

**Effect of Label Refinement.** To verify how the label refinement affects the performance, we do not use SAM to refine the pseudo-labels during training (**w/o Refine**); we perform prediction refinement in the final result (**Post-Refine**). Notably, we observed a significant drop in accuracy when SAM refinement was omitted during training. This suggests that SAM is crucial in enhancing the model's performance. However, we also noticed that post-refinement did not contribute to the performance. This can be attributed to the fact that the trained networks already produced semantic-consistent regions similar to those obtained with SAM refinement.

**Effect of Latent Space Consistency Regularization.** We utilize two main features to evaluate the latent space consistency regularization: the **CLIP** feature and the **CLIP+SAM** feature. In addition, we have experimented without any Latent Space Regularization (**None**). Results demonstrate that our full method incorporating the SAM feature performs best. On the other hand, we have observed that the CLIP feature alone is less effective in our task, primarily because the feature space it provides is not specifically designed for segmentation purposes.

**Effect of Prediction Consistency Regularization.** We conducted five configurations to examine the effects of prediction consistency regularization: 1) **w/o CT**: In this experiment, the 2D and 3D networks were not allowed to cross-train each other in the second training stage, *i.e.*, the 2D network's labels were randomly switched between CLIP and 2D prediction. Meanwhile, the 3D network's labels were randomly switched between CLIP and 3D prediction; 2) **w/o SCT**: This experiment eliminates self- and cross-training in 2D and 3D networks, *i.e.*, 2D and 3D network labels are derived solely from CLIP; 3) **w/o CLIP**: This experiment excludes the usage of CLIP's labels during the second training stage, *i.e.*, both the 2D and 3D network's labels were randomly switched between 2D and 3D prediction; And 4) **w 2D**, 5) **w 3D**: we employ either 2D or 3D network prediction to

synchrony train 2D and 3D networks, respectively. Experiments show that SAM refinement promotes pseudo-label quality. Besides, The experimental results indicate that cross-training, self-training, and label-switching effectively improve performance. Importantly, we observed that the method fails when CLIP's supervision is absent during the second training stage. This failure can be attributed to the tendency of the 2D and 3D networks to "co-corrupt" each other with extremely noisy labels.

**Effect of Images Numbers.** We also show the performance when restricting image numbers to 8, 16, and 32, respectively. However, we observe that the performances are comparable. Therefore, for efficient training, we set the image number to 16.

## 5 Conclusion

In this work, we study how vision foundation models enable networks to comprehend 2D and 3D environments without relying on labelled data. To address noisy pseudo-labels, we introduce a cross-modality noisy supervision method to simultaneously train 2D and 3D networks, which involves the label refinement strategy, prediction consistency regularization, and latent space consistency regularization. Our method achieves state-of-the-art performance in understanding open worlds, as demonstrated by extensive experiments on indoor and outdoor datasets.

## 6 Acknowledgements

Yuexin Ma was partially supported by NSFC (No.62206173), MoE Key Laboratory of Intelligent Perception and Human-Machine Collaboration (ShanghaiTech University), and Shanghai Frontiers Science Center of Human-centered Artificial Intelligence (ShangHAI). Tongliang Liu was partially supported by the following Australian Research Council projects: FT220100318, DP220102121, LP220100527, LP220200949, and IC190100031.

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
