# OpenReview forum: "Towards Label-free Scene Understanding by Vision Foundation Models"
_NeurIPS.cc/2023/Conference — NeurIPS 2023 poster_

### Official Review · Reviewer_inkR · 2023-07-02

**Soundness:** 3 good
**Presentation:** 3 good
**Contribution:** 2 fair
**Rating:** 7
**Confidence:** 4

**Summary:**

The paper proposes a method that utilizes the SAM and CLIP (Contrastive Language-Image Pretraining) models for understanding the 2D and 3D world in a label-free manner. The authors suggest a two-step process to improve the results obtained from CLIP.

In the first step, the authors generate a noisy output from the CLIP model, which is primarily designed for semantic tasks. This output may contain imperfections or inaccuracies due to the inherent limitations of CLIP.To refine the output and obtain more visually appealing semantic results, the authors employ a SAM network in the second step. The SAM network is designed to enhance the quality of the semantic results generated by CLIP.

Furthermore, the paper introduces a Cross-modality Noise Supervision module, which aims to optimize the training of both the 2D and 3D models simultaneously.
The authors demonstrate promising results on datasets such as ScanNet and nuScenes. These datasets are commonly used in the field of computer vision and provide challenging scenarios for understanding the 2D and 3D world. The encouraging results obtained from these datasets suggest the effectiveness of the proposed approach in understanding and representing the visual and semantic aspects of the real world.

**Strengths:**

Strength:

1. Paper pushes the current boundary of training models with label free training data, using pre-existing models which have shown good performance on zero-shot tasks.
2. Proposes to seamlessly refine labels predicted by CLIP model to achieve good results in Indoor Dataset.
3. Paper for most of the part is well written and has a definite flow to it.
4. Beats existing method such MaskCLIP, CLIP2scene models comprehensively to get state-of-arts results.

**Weaknesses:**

1. Missing Ablation: Prediction Consistency Regularization, the second stage where there is random switching of pseudo labels between 2D, 3D and CLIP based intermediate outputs, is not well explained.

2. Missing Ablation: Ablation study which shows what is the impact of this random switching with and without. We also need to see what if we only employ one of the three pseudo labels and not all three, what impact does that have. Seeing the impact of each individual is necessary to comment about its efficacy.

3. Effect of Latent Space Consistency Regularization: There is only marginal improvement seen using SAM features to guide the feature of intermediate 2D/3D encoder features. Do the authors have intermediate results(qualitative), which justifies how SAM features actually help the 2D and 3D features ?

Typos:
1. Line no 169: is the doc product operation. I presume this is supposed to be dot product.
2. In Fig:2, How do we achieve the Refined CLIP Pseudo Label and how is this different from Refined 2D Pseudo labels. Please add further information for this.

**Questions:**

1. Could the authors provide intermediate feature space visualization which shows comparative results with and without latent space regularization ? Since from the ablation experiments provided there seems to be marginal improvement in mIoU numbers both in the ScanNet and nuScenes dataset.

**Limitations:**

1. The method proposed assumes SAM to be perfect in sense of providing meaning masks and does not train the SAM model. But fails to give examples or quote scenarios where this might fail, considering SAM outputs a mask which is not meaningful or helping to noisy CLIP based labels.

---

> ### Author Rebuttal · Authors · 2023-08-09
>
> We thank Reviewer inkR for devoting time to this review and providing valuable comments.
>
> ---
> > ***Weaknesses 1:** "Random switching of pseudo labels is not well explained."*
>
> **A:** Thanks! We would like to explain how we conducted ablation experiments about label random switching. The ablation experiment settings are in the paragraph "Effect of Prediction Consistency Regularization" (lines 296-304), and we have elaborated this aspect in a more straightforward way in the revised version:
>
> 1) In our full method, both the 2D and 3D network's labels were randomly switched among CLIP, 2D and 3D prediction.
> 2) We experimented with the setting "w/o CT". In this experiment, the 2D and 3D networks were not allowed to cross-train each other in the second training stage, i.e., the 2D network's labels were randomly switched between CLIP and 2D prediction. Meanwhile, the 3D network's labels were randomly switched between CLIP and 3D prediction.
> 3) We experimented "w/o SCT". This experiment eliminates self- and cross-training in 2D and 3D networks, i.e., 2D and 3D network labels are derived solely from CLIP.
> 4) The experiment "w/o CLIP", this experiment excludes the usage of CLIP's labels during the second training stage, i.e., both the 2D and 3D network's labels were randomly switched between 2D and 3D prediction.
>
> As shown in Table 2, our full method achieves the best performance. We also observe that the method fails when CLIP's supervision is absent during the second training stage. This failure can be attributed to the tendency of the 2D and 3D networks to "co-corrupt" each other with extremely noisy labels.
>
> ---
> > ***Weaknesses 2:** "Missing Ablation: Ablation study which shows what is the impact of this random switching with and without."*
>
> **A:** Thanks for your suggestions! We additionally conduct the following ablation experiments to see the impact of random switching with and without.
> 1) "W CLIP" (the same as "w/o SCT"): employing only CLIP's prediction to train 2D and 3D networks. The mIoU is 24.7 and 30.6 in 2D and 3D, respectively.
> 2) "W 2D": employing only 2D network's prediction to train 2D and 3D networks. The mIoU is 6.9 and 8.8 in 2D and 3D, respectively.
> 3) "W 3D": employing only 3D network's prediction to train 2D and 3D networks. The mIoU is 4.8 and 6.4 in 2D and 3D, respectively.
>
>
> |Method|ScanNet 2D|ScanNet 3D|
> |-|:-:|:-:
> |W CLIP|24.7|30.6
> |W 2D|6.9|8.8
> |W 3D|4.8|6.4
> |Full method|28.4|33.5
>
> The experiments show the effectiveness of employing the random switching strategy, and the 2D and 3D tend to corrupt if without the guidance of CLIP's pseudo labels. We have supplemented the above experiments in the revised version. Thanks!
>
> ---
> > ***Weaknesses 3:** "Effect of Latent Space Consistency Regularization."*
>
> **A:** - The performance gain in Latent Space Consistency Regularization seems marginal because it is conducted on our full method. Therefore, we conduct the following ablation experiments, which may verify the effectiveness better. Firstly, we impose the baseline method (lines 272) self-training with the CLIP pseudo-labels, termed "baseline + CLIP". Secondly, building upon "baseline + CLIP", we additionally impose Latent Space Consistency Regularization, termed "baseline + CLIP + LSCR". The improvements from "baseline + CLIP" to "baseline + CLIP + LSCR" are 2.1 and 4.9 mIoU in ScanNet 2D and 3D datasets, respectively, which is more prominent.
> |Method|ScanNet 2D|ScanNet 3D|
> |-|:-:|:-:
> |baseline|17.3|14.2
> |baseline + CLIP|20.4|23.8
> |baseline + CLIP + LSCR|22.5|28.7
> |Full method|28.4|33.5
>
> ---
> > ***Typos 1:** "Line no 169: is the doc product operation. I presume this is supposed to be dot product."*
>
> **A:** Thanks for pointing out the typos, we have fixed it in the revised version.
>
> ---
> > ***Typos 2:** "How do we achieve the Refined CLIP Pseudo Label and how is this different from Refined 2D Pseudo labels."*
>
> **A:** Thanks for your suggestions! Please check our improved figure in the supplement PDF file. Specifically, we detailed the process of CLIP's pseudo-labeling and label refinement (on the right-hand side). We hope it is better for understanding our method. Please let us know if there is anything unclear.
>
>
> ---
> > ***Questions 1:** "Could the authors provide intermediate feature space visualization which shows comparative results with and without latent space regularization ?"*
>
>
> **A:** Thanks for your valuable suggestion! As shown in the supplement pdf file, we visualize the feature space with (w LSCR) or without (w/o LSCR) latent space regularization by using T-SNE (t-distributed Stochastic Neighbor Embedding) [P1]. T-SNE is an unsupervised non-linear dimensionality reduction technique for data exploration and visualizing high-dimensional data. By our latent space regularization, for those pixels/points with the same semantics (the same colour), their feature space tends to be more compact. Meanwhile, for those pixels/points with different semantics (the different colours), their feature space tends to be more distinguishable.
>
>
> ---
> > ***Limitations 1:** "The method proposed assumes SAM to be perfect in the sense of providing meaning masks."*
>
> **A:** Good points! Indeed, the label refinement module heavily relies on the SAM's prediction, and SAM is not always perfect. We observe it may fail in bad weather conditions, where the scenes are burr and caliginous. However, the label refinement module is just one of the components of our Cross-modality Noisy Supervision method. The other two modules, Latent Space and Prediction Consistency Regularization, are able to further mitigate error propagation and prevent the networks from overfitting the noisy labels (lines 53-60). The Ablation experiments in Table 2 also verify the effectiveness of these modules. We have supplemented the discussion in the revised version. Thanks!
>
> ---
> **References:**
> - [P1] L. Van der Maaten and G. Hinton. Visualizing High-Dimensional Data Using t-SNE. JMLR, 2008.

---

> > ### Comment · Reviewer_inkR · 2023-08-16
> > **Reply to Authors**
> >
> > Thank authors for providing supporting answers and results. I think the authors have almost answered and provided results for most of my queries and I hope they would add these results in the final version of the paper.
> >
> > I had one query though the t-SNE visualization and statement given "By our latent space regularization, for those pixels/points with the same semantics (the same color), their feature space tends to be more compact. Meanwhile, for those pixels/points with different semantics (the different colors), their feature space tends to be more distinguishable." does not quite hold good for all the visualization as presented in the result-pdf file.
> >
> > Effect on latent feature space still needs some further investigation to evaluate its impact and identify probable weakness in the current approach. Never the less I will change my rating to Accept from Weak Accept after author's rebuttal.

---

> > > ### Author Response · Authors · 2023-08-17
> > > **Authors' Response to Reviewer inkR**
> > >
> > > Thanks so much for donating your time to our paper. As suggested, we will add these results in the final version. Besides, we have thought about multiple potential ways to better verify the effectiveness of the latent space feature regularization:
> > >
> > > - We could apply linear probing on the latent space feature, i.e., using a few labelled data to fine-tune the network and compare the improvements to that without latent space feature regularization.
> > > - We could unsupervised cluster the feature and qualitatively evaluate the performance using ground-truth labels.
> > > - We could set different anchor points and visualize the similarity map based on the similarity between the anchor point feature and all other point features.
> > >
> > > We will investigate and discuss the probable weakness in the final version. Thanks again for your valuable comments and positive feedback!

---

### Official Review · Reviewer_rM41 · 2023-07-03

**Soundness:** 3 good
**Presentation:** 2 fair
**Contribution:** 2 fair
**Rating:** 4
**Confidence:** 4

**Summary:**

This work employs 2D segmentation foundation models to perform semantic segmentation for both 2D and 3D indoor scenes. The clip model provides a general understanding of the semantic content in each image, while the SAM model generates precise segmentation masks. By integrating these two models, this study demonstrates improved quantitative results on two datasets.

**Strengths:**

1. The significant enhancement in segmentation results is achieved by leveraging two distinctively trained 2D segmentation models, SAM and Clip, surpassing the previous approach that solely utilized the Clip model.

2. It is reasonable to anticipate improved segmentation outcomes by combining two powerful models that provide both semantic understanding and precise mask contours. The merging of diverse masks through pooling is straightforward and appears to be the most effective element contributing to the performance gain.

**Weaknesses:**

1. Writing: The authors are encouraged to provide further clarification on the "calibration matrix," as it serves as the initial step in connecting 2D pixels and 3D points. How was this connection established without knowledge of the depth of each pixel? Additionally, could you please clarify the meaning of "Image Number" mentioned in line 305 and indicate the corresponding table or figure for reference? A similar question arises regarding the ablation study mentioned in line 290.

2. Clarity:  In the abstract, the authors claim that the proposed method outperforms the state-of-the-art by a significant margin, but the method's name is not mentioned. When the term "calibration matrix" is first introduced in line 51, could you please provide a clear explanation? Does the proposed method predict both 2D and 3D semantic label maps? In Figure 2, which model requires training/finetuning and which one is frozen? How is the SAM prompted to predict the segmentation maps?

3. As a straightforward baseline, the authors could utilize Clip to generate coarse semantic masks and merge these masks with SAM segmentation results. It is recommended to present the performance in Table 2 and the corresponding figure. The suggested experiments should be evaluated on the large-scale Scannet dataset. Furthermore, if the 2D foundation models already perform well, the motivation behind the "Prediction Consistency Regularization" is not clear. Why is there a need to fine-tune your own model?

4. Although the demonstrated segmentation results are close to the ground truth, the mIoU scores for both 2D and 3D remain below 35.

5. The improvement achieved through "Latent Space Consistency Regularization" appears to be minimal for both 2D and 3D segmentation, despite being listed as an important contribution.

6. Please include citations for NeRF-based and mesh-based scene segmentation:

[1] Atlas: End-to-end 3d scene reconstruction from posed images (supervised and unsupervised)
[2] Decomposing nerf for editing via feature field distillation
[3] Nerf-sos: Any-view self-supervised object segmentation on complex scenes
[4] Unsupervised Multi-View Object Segmentation Using Radiance Field Propagation

**Questions:**

As the reviewer's question mainly lies in the clarity and writing, please refer to the "Weakness" section.

**Limitations:**

Please refer to the "Weakness" section.

---

> ### Author Rebuttal · Authors · 2023-08-09
>
> We thank Reviewer rM41 for devoting time to this review and providing valuable comments.
>
> ---
> > ***Weaknesses 1:** "Writing: The authors are encouraged to provide further clarification on the "calibration matrix," as it serves as the initial step in connecting 2D pixels and 3D points."*
>
> **A:** Thanks for your detailed comments! We intend to express that we transfer the pseudo-label from 2D image pixels to 3D points using pixel-point correspondence. Pixel-point correspondence is obtained by direct projection between the 2D images and the 3D point cloud, where the pixel depth and camera poses are provided in the dataset. We have improved the presentation in the revised version. Thanks!
>
>
> ---
> > ***Weaknesses 2:** "2. Clarity: In the abstract, the authors claim that the proposed method outperforms the state-of-the-art by a significant margin, but the method's name is not mentioned. When the term "calibration matrix" is first introduced in line 51, could you please provide a clear explanation? Does the proposed method predict both 2D and 3D semantic label maps? In Figure 2, which model requires training/finetuning, and which one is frozen? How is the SAM prompted to predict the segmentation maps?"*
>
> **A:**
> - Thanks for your comments. We will acknowledge the state-of-the-art CLIP2Scene in the abstract.
> - The calibration matrix here means the pixel-point correspondence that transfers pseudo-label from 2D image pixels to 3D points. We have improved the presentation in the revised version.
> - Our method is designed for label-free 2D and 3D scene understanding, so it predicts both 2D and 3D semantic label maps (lines 16 and 76).
> - We have improved Figure 2. As shown in the supplement pdf file, 2D and 3D networks are needed for training, while SAM and CLIP are frozen during training.
> - Following the official implementation of SAM, we use the function "SamAutomaticMaskGenerator" to produce the masks. Basically, it first uniform samples the seed points in an image, then cluster those pixels with similar features to the seed points. As a result, each cluster can be regarded as a mask for the images. We have supplemented more details in the revised version. Thanks!
>
> ---
> > ***Weaknesses 3:** "It is recommended to present the performance in Table 2 and the corresponding figure. The suggested experiments should be evaluated on the large-scale Scannet dataset. Furthermore, if the 2D foundation models already perform well, the motivation behind the "Prediction Consistency Regularization" is not clear. Why is there a need to fine-tune your own model?"*
>
> **A:** Thanks for your comments. We would like to clarify that we have conducted such experiments in the paper, termed "Ours w/o CNS". The qualitative and quantitative results are shown in Figures 3, 4, and Table 2. And more 2D and 3D qualitative results are shown in the supplementary materials. Note that all the experiments are evaluated on 25K 2D images and 1513 3D scans in the ScanNet dataset (lines 239), which is the most comprehensive dataset to our knowledge. Besides, we analyze the results in the ablation section (lines 275). As shown in Figures 3, 4, and Table 2, "Ours w/o CNS" achieves sub-optimal performance. The merged CLIP and SAM results are unsatisfactory and still suffer from noise prediction. The motivation behind the "Prediction Consistency Regularization" is that different networks have merits in different aspects. Co-training [P1] the different networks helps to mitigate error propagation and prevent the networks from overfitting the noisy labels (lines 53). We have clarified the above details in the revised version. Thanks!
>
> ---
> > ***Weaknesses 4:** "4.Although the demonstrated segmentation results are close to the ground truth, the mIoU scores for both 2D and 3D remain below 35."*
>
> **A:** Label-free scene understanding is a very challenging task. We show some representative cases and hope to inspire the research community. Besides, there are also some failure cases in the supplementary materials, especially in the nuScenes dataset. Thanks!
>
> ---
> > ***Weaknesses 5:** "The improvement achieved through "Latent Space Consistency Regularization" appears to be minimal for both 2D and 3D segmentation."*
>
> **A:**
> - The performance gain in Latent Space Consistency Regularization seems marginal because it is conducted on our full method. Therefore, we conduct the following ablation experiments, which may verify the effectiveness better. Firstly, we impose the baseline method (lines 272) self-training with the CLIP pseudo-labels, termed "baseline + CLIP". Secondly, building upon "baseline + CLIP", we additionally impose Latent Space Consistency Regularization, termed "baseline + CLIP + LSCR". The improvements from "baseline + CLIP" to "baseline + CLIP + LSCR" are 2.1 and 4.9 mIoU in ScanNet 2D and 3D datasets, respectively, which is more prominent.
>
> |Method|ScanNet 2D|ScanNet 3D|
> |-|:-:|:-:
> |baseline|17.3|14.2
> |baseline + CLIP|20.4|23.8
> |baseline + CLIP + LSCR|22.5|28.7
> |Full method|28.4|33.5
>
> - Besides, as shown in the supplement pdf file, we visualize the feature space with (w LSCR) or without (w/o LSCR) latent space regularization by using T-SNE (t-distributed Stochastic Neighbor Embedding) [P1]. T-SNE is an unsupervised non-linear dimensionality reduction technique for data exploration and visualizing high-dimensional data. By our latent space regularization, for those pixels/points with the same semantics (the same colour), their feature space tends to be more compact. Meanwhile, for those pixels/points with different semantics (the different colours), their feature space tends to be more distinguishable.
>
> ---
> > ***Weaknesses 6:** "Please include citations for NeRF-based and mesh-based scene segmentation."*
>
> **A:** We will investigate and discuss the methods in the revised version. Thanks!
>
> ---
> **References:**
> - [P1] Co-teaching: Robust Training of Deep Neural Networks with Extremely Noisy Labels

---

> > ### Comment · Reviewer_rM41 · 2023-08-21
> > **Thanks for the authors' response**
> >
> > After reviewing the authors' response to the posted inquiries, additional crucial details regarding the methodology have been incorporated, along with an explanation concerning the effectiveness of the "Latent Space Consistency Regularization." My anticipated score improvement is grounded in the following reasons:
> >
> > 1. The proposed approach is intriguing, employing two foundational models for scene comprehension, resulting in commendable outcomes.
> > 2. The seemingly straightforward "Label Refinement" + Post-Refine technique exhibits substantial power in enhancing overall accuracy.
> > 3. It's advisable to incorporate more representative outcomes within the main paper, ensuring quantitative figures align with qualitative findings: the overall mIoU is still far from the supervised conterpart, but the visualizations in the main draft seems perfect.
> > 4. While the integration of diverse model insights can undoubtedly heighten accuracy, it's essential for the authors to perform runtime and memory consumption analyses as outlined in Table 1, ensuring equitable comparisons.

---

> > > ### Author Response · Authors · 2023-08-21
> > > **Authors' Response to Reviewer rM41**
> > >
> > > We appreciate your acknowledgement of the interest in our method and the effectiveness of the Label Refinement module. Additionally, we are committed to enhancing our paper based on your suggestions:
> > >
> > > 1. A major contribution of the paper is that we study how vision foundation models enable networks to comprehend 2D and 3D environments without relying on labelled data. Thanks for your acknowledgement.
> > >
> > > 2. The Label Refinement is an important module in our proposed Cross-modality Noisy Supervision framework, which is one of the major contributions of our method. Thanks for your acknowledgement.
> > >
> > > 3. Due to space constraints, we have included more qualitative results, including failure cases, in the supplementary materials. In the final version, we intend to present a more comprehensive selection of representative results within the main paper. Thanks for your understanding.
> > >
> > > 4. We would like to emphasize that our method exhibits comparable runtime and memory consumption to other state-of-the-art approaches during the inference stage. This similarity arises from the shared utilization of 2D and 3D backbones with other methods. It's important to note that the vision foundation models (CLIP and SAM) are exclusively employed during the training phase. In our final version, we will provide further elaboration on these technical details. Thank you for highlighting this aspect.
> > >
> > > Once again, we express our gratitude for your feedback. We hope that our response addresses all of your concerns. If you identify any additional improvements within our paper, please do not hesitate to share them with us.

---

> ### Author Response · Authors · 2023-08-18
> **Anticipating the response from Reviewer rM41**
>
> Thank you once again for your invaluable contribution to our paper. We hope that the response provided above adequately addresses your concerns. If you find any aspects of the paper unclear or have further questions, please don't hesitate to share them with us. We are committed to resolving any uncertainties and ensuring the clarity and quality of our work. Your response is highly appreciated. Thank you!

---

### Official Review · Reviewer_Mxk9 · 2023-07-03

**Soundness:** 4 excellent
**Presentation:** 3 good
**Contribution:** 4 excellent
**Rating:** 6
**Confidence:** 5

**Summary:**

This paper proposes using SAM to improve the quality of segmentation masks generated by CLIPs. The refined masks are then used as pseudo-labels to train networks that excel at segmentation tasks, following a similar training process to MaskCLIP+. Based on this idea, a two-stage self-training process is proposed to simultaneously train 2D and 3D networks with SAM-refined masks and self-generated masks iteratively. To mitigate mask noise, the authors further propose latent space consistency regularization to transfer knowledge from the SAM feature space to segmentation networks. The final trained segmentation networks demonstrate impressive performance on both 2D and 3D segmentation datasets.

**Strengths:**

(1) The overall method is both simple and effective. Although the authors use SAM trained with labeled data, which means it is not strictly label-free, I believe that it is acceptable given the performance improvement and contribution to the community.

(2) The performance is impressively prefect.

(3) The ablation study proves the gains of the main contributions of the paper.

(4) The paper is well-written and easy to read and understand except for some minor issues.


**Weaknesses:**

(1) In this paper, the authors chose DeeplabV3 as the 2D backbone and MinkowskiNet34 as the 3D backbone. However, previous works such as MaskCLIP used DeeplabV2 as the 2D backbone and CLIP2Scene used MinkowskiNet14 as the 3D backbone. It is obviously that the backbones used in this paper have better performance, so it may not be entirely fair to compare them with previous works.

(2) I'm a bit confused about the details of the Label Refinement by SAM. SAM often outputs very fine-grained masks and produces multi-level masks, while CLIP also often outputs masks with a lot of discrete noise points. How to handle these situations? Can the author provide a simple pseudo-code to describe this part of the processing?

(3) I think the baseline selected by this paper is not appropriate. While the main contributions of this paper are the improvement of pseudo-labeling quality and more effective training strategies, the baseline should be a basic method that incorporates both of these points. For example, I think that for 2D scenes, MaskCLIP+ is a better baseline than CLIP.

(4) I think the presentation of Table 2 could be improved. Except for the experimental settings in the first row, the configurations in the other rows are abridged versions of the full configuration in a specific structure. To better present the impact of each structure on the final configuration, I suggest comparing the results of each ablation experiment with the performance drop of the full configuration results.

(5) MaskCLIP++ -> MaskCLIP+.


**Questions:**

Please refer to Weaknesses.

**Limitations:**

Yes, the authors adequately addressed the limitations.

---

> ### Author Rebuttal · Authors · 2023-08-09
>
> We thank Reviewer Mxk9 for devoting time to this review and providing valuable comments.
>
> ---
> > ***Weaknesses 1:** "the authors chose DeeplabV3 as the 2D backbone and MinkowskiNet34 as the 3D backbone. However, previous works such as MaskCLIP used DeeplabV2 as the 2D backbone and CLIP2Scene used MinkowskiNet14 as the 3D backbone."*
>
> **A:** Thanks for your detailed comments. We would like to claim that we re-implement MaskCLIP for comparison. Therefore, the re-implement MaskCLIP shares the same 2D backbone (DeeplabV3) as well as the same data augmentations with our method. As for the comparison with CLIP2Scene, we conduct the experiment that adapts MinkowskiNet14 as the 3D backbone for our method. We achieve 27.6, 32.3, and 25.2 mIoU in ScanNet 2D, 3D, and nuScenes datasets, which still significantly outperform CLIP2Scene. We have supplemented more experiment details in the revised version. Thanks!
>
> |Method|ScanNet 2D|ScanNet 3D|nuScenes
> |-|:-:|:-:|:-:
> |CLIP2Scene|23.7|25.6|20.8
> |Ours (MinkowskiNet14)|27.6|32.3|25.2
> |Ours|28.4|33.5|26.8
>
> ---
> > ***Weaknesses 2:** "I'm a bit confused about the details of the Label Refinement by SAM"*
>
> **A:** We follow the official implementation of SAM that produces object masks and remove the redundant masks. Next, we enumerate each mask and adjust the pseudo-labels to be identical within the same mask. Specifically, we adopt the max voting strategy that unifies all pixels' labels to be the semantic labels that most pixels belong to. Our supplementary materials provide the source code in the function "label_refinement_sam" of the file: "codes/pretrain/lightning_trainer.py". By the way, we visualize the qualitative results of the Label Refinement by SAM in Figures 2 and 3 and the Figures in the supplementary materials, where "Ours w/o CNS" indicates the results of CLIP's label Refinement by SAM. Besides, we present the following pseudo-code for better understanding.
>
>       Algorithm: label_refinement_sam
>
>       Input: pseudo-labels (H * W), masks_sams (N * H * W)
>
>       (H and W indicate the height and width of an image. N is the number of masks. A mask consists of 0 or 1.)
>
>       Process:
>
>       for i, mask in enumerate(masks_sam):
>
>            pseudo-labels[mask] = argmax(pseudo-labels[mask])
>
>       Return: pseudo-labels
>
> > ***Weaknesses 3:** "I think the baseline selected by this paper is not appropriate."*
>
> **A:** Thanks for your suggestion. We have adopted MaskCLIP+ as the 2D baseline in the revised version.
>
> > ***Weaknesses 4:** "I think the presentation of Table 2 could be improved."*
>
> **A:** Good points! We will follow your suggestion in the revised version. Thanks!
>
> > ***Weaknesses 5:** "MaskCLIP++ -> MaskCLIP+."*
>
> **A:** Thanks for pointing out the typo. We have fixed it in the revised version.

---

> > ### Comment · Reviewer_Mxk9 · 2023-08-17
> >
> > The rebuttal solves most of my concerns. I keep my rating as weak accept. I wish that the authors correct the minor errors and further improve quality of the paper in the revised version.

---

> > > ### Author Response · Authors · 2023-08-17
> > > **Authors' Response to Reviewer Mxk9**
> > >
> > > We thank Reviewer Mxk9 for participating in the Author-Reviewer discussion session and providing positive feedback. As suggested, we will revise the manuscript rigorously based on the reviewers' comments.
> > >
> > > ---
> > > Last but not least, we thank Reviewer Mxk9 again for the time and effort devoted and the valuable comments drawn during this review.

---

### Official Review · Reviewer_9W6n · 2023-07-07

**Soundness:** 3 good
**Presentation:** 3 good
**Contribution:** 3 good
**Rating:** 6
**Confidence:** 4

**Summary:**

The paper proposes a new method for jointly learning 2D and 3D segmentation models that can be queried with open-world text queries. The paper's main contribution is a new Cross-modality Noisy Supervision to reduce the influence of noise on the predictions. The proposed method uses a SAM model to predict masks for every pixel and then uses those noises to regularize CLIP's noisy dense pixel features. The results shown in the paper show that the proposed method outperforms baselines in both 2D and 3D segmentation. I have a couple of questions and suggestions to improve the paper and I am willing to increase my score if these are sufficiently addressed in the rebutal.

**Strengths:**

The paper is overall well-written and mostly easy to follow (apart from the mathematical notation, see comments below). The paper provides a pretty comprehensive overview of the literature up until CVPR 2023, and I also liked the qualitative results and Figure 1, which was informative.

**Weaknesses:**

I have some suggestions (in no particular order) to improve the paper:

1. I take some issue with the claim that the method is the first to combine SAM and CLIP for label-free scene understanding. Multiple different methods have combined 2D segmentation models with CLIP, and as these systems are built modularity, they have already been able to switch out their segmenters for SAM. Two papers that come to mind for this are OvSeg (2D segmentation) and ClipFusion (3D segmentation). I suggest that this claim be softened.

2. Figure 2 is very non-intuitive in my opinion and I did not fully understand the flow of information even after multiple reads. Maybe it would be good to improve this figure.

3. Section 3.2 seems to be overly mathy and the notation is extremely convoluted. Consider cleaning up the super/sub scripts, and I also have a few additional comments:
* Equations 2 and 3 take a lot of space do not add anything to my understanding of the method. I suggest removing them.
* It is not necessary to define the range of each variable.
* Equations 5/6 are extremely convoluted. Consider simpler notation.
* In equation 1, I think there is an error, and the dot product should be between c_i^T, t_l

4. Regarding the results in Table 1, the OpenScene results seem to be much lower compared to the ones reported in the original paper. How can this be explained? Was the baseline retrained for this paper?





**Questions:**

I have a few clarification questions as well:

1. Line 159: What is meant by pixel-point calibration? Are the camera poses not known? Is there an additional step required apart from simple projection? If not, this section can be shortened in my opinion.

2. Line 160: What is meant by the domain gap here? And why does this matter in the first place? As I see it, the method tries to align the features from the 3D network to the 2D CLIP/SAM space, and the domain gap should not matter here, right?

3. Line 209: In the current method, you use cosine similarity to force 2D/3D network outputs to be similar to the SAM feature output. Is this objective not in contrast to enforcing similarity between CLIP features and the 3D/3D network outputs? Especially when looking at the results in Table 2, this specific regularization does not seem to add a lot in terms of performance (Latent Space Consistency Regularization)

**Limitations:**

The paper does not discuss limitations and broader impact in the paper.

---

> ### Author Rebuttal · Authors · 2023-08-09
>
> We thank Reviewer 9W6n for devoting time to this review and providing valuable comments.
>
> ---
> > ***Weaknesses 1:** "The claim that the method is the first to combine SAM and CLIP."*
>
> **A:** Thanks for your suggestion! Considering previous methods can easily replace their 2D network with SAM, we will modify the unrigorous claim "the first work to combine SAM and CLIP". Instead, we would like to mention that we study how CLIP and SAM enable networks to comprehend 2D and 3D environments without relying on labelled data.
>
> ---
> > ***Weaknesses 2:** "Figure 2 is very non-intuitive."*
>
> **A:** Thanks for your valuable advice! As shown in the supplement PDF file, we have improved Figure 2 to demonstrate the framework better. Specifically, we visualize the intermedia results of network predictions. Besides, we detailed the process of CLIP's pseudo-labeling and label refinement (on the right-hand side). Please let us know if there is anything unclear.
>
> ---
> > ***Weaknesses 3:** "Section 3.2 seems to be overly mathy and the notation is extremely convoluted."*
>
> **A:** We will correct Equations 1, remove Equations 2 and 3, and clean the variable range (super/sub-scripts). We will simplify the loss function (Equations 5/6). For example Equations 5, $\mathcal{L}_{s1} = \sum\_{i} CE(x_i, P^*\_{x_i}) + CE(p_i, P^*\_{p_i})$, where $CE(x, y) =- \log\frac{\exp(\psi(\theta\_s(x), t\_{y}))}{\sum\_{l}\exp(\psi(\theta\_s(x), t\_{l}))}$. Thanks!
>
> ---
> > ***Weaknesses 4:** "Regarding the results in Table 1."*
>
> **A:** OpenScene adapts the OpenSeg and LSeg as the 2D backbone, where OpenSeg and LSeg are trained on labelled images and significantly outperform our 2D backbone MaskCLIP. Therefore, for a fair comparison in the label-free setting, we replace OpenScene's 2D backbone with MaskCLIP, which is without training on any labelled images. To this end, OpenScene achieves much lower performance than that reported in the original paper.
>
> ---
> > ***Questions 1:** "Line 159: What is meant by pixel-point calibration? Are the camera poses not known? Is there an additional step required apart from simple projection?"*
>
> **A:** Pixel-point calibration means pixel-point correspondence by 2D image and 3D point cloud calibration. Note that the dataset already provides the camera poses, so we obtain the pixel-point correspondence via simple projection. We have fixed the ambiguous expression in the revised version. Thanks!
>
> ---
> > ***Questions 2:** "Line 160: What is meant by the domain gap here? And why does this matter in the first place?"*
>
> **A:** The domain gap means that 2D images and 3D point clouds have different representations. Therefore, the vision foundation models, which are trained on 2D images, could not be directly used to process 3D point clouds. To this end, we transfer the vision foundation models' knowledge from 2D images to 3D point clouds via dense pixel-point correspondence. We have improved the presentation in the revised version. Thanks!
>
> ---
> > ***Questions 3:** "Line 209: You use cosine similarity to force 2D/3D network outputs to be similar to the SAM feature output. Is this objective not in contrast to enforcing similarity between CLIP features and the 3D/3D network outputs?"*
>
> **A:** Thanks for the comment.
> - Good question! We have encountered this issue and found that the optimization conflict reduces performance. To solve this issue, we adopt a simple yet effective solution in our paper. Specifically, we use different linear mapping layers in Latent Space Consistency Regularization and Prediction Consistency Regularization. That is, the 2D and 3D features are respectively mapped to other feature spaces by $\theta_{f}$ and $\varphi_{f}$ (lines 222)before aligning with the SAM feature. Meanwhile, 2D and 3D features are respectively mapped by $\theta_s$ and $\varphi_s$ (lines 193) before aligning with the CLIP's text embedding. Therefore, these two optimizations are not directly conflict with each other.
>
> - The performance gain in Latent Space Consistency Regularization seems marginal because it is conducted on our full method. Therefore, we conduct the following ablation experiments, which may verify the effectiveness better. Firstly, we impose the baseline method (lines 272) self-training with the CLIP pseudo-labels, termed "baseline + CLIP". Secondly, building upon "baseline + CLIP", we additionally impose Latent Space Consistency Regularization, termed "baseline + CLIP + LSCR". The improvements from "baseline + CLIP" to "baseline + CLIP + LSCR" are 2.1 and 4.9 mIoU in ScanNet 2D and 3D datasets, respectively, which is more prominent.
> |Method|ScanNet 2D|ScanNet 3D|
> |-|:-:|:-:
> |baseline|17.3|14.2
> |baseline + CLIP|20.4|23.8
> |baseline + CLIP + LSCR|22.5|28.7
> |Full method|28.4|33.5
>
> - Besides, as shown in the supplement pdf file, we visualize the feature space with (w LSCR) or without (w/o LSCR) latent space regularization by using T-SNE (t-distributed Stochastic Neighbor Embedding) [P1]. T-SNE is an unsupervised non-linear dimensionality reduction technique for data exploration and visualizing high-dimensional data. By our latent space regularization, for those pixels/points with the same semantics (the same colour), their feature space tends to be more compact. Meanwhile, for those pixels/points with different semantics (the different colours), their feature space tends to be more distinguishable.
>
> ---
> > ***Limitations:** "The paper does not discuss limitations and broader impact in the paper."*
>
> **A:** Limited by space, we discussed the limitation and future work in supplementary materials. As for the broader impacts, we believe our method is meaningful for the artificial general intelligence field because it bridges the vision and language, percept real-world environments without relying on any labelled data. We hope our work will inspire and motivate further research in this area.
>
> ---
> **References:**
> - [P1] L. Van der Maaten and G. Hinton. Visualizing High-Dimensional Data Using t-SNE. JMLR, 2008.

---

> > ### Comment · Reviewer_9W6n · 2023-08-17
> >
> > I want to thank the reviewers for their in-depth explanations and clarifications. They have effectively addressed most of my concerns, and I will raise my score to a weak accept. I think that this paper is timely and addresses an interesting issue.

---

> > > ### Author Response · Authors · 2023-08-18
> > > **Authors' Response to Reviewer 9W6n**
> > >
> > > We thank Reviewer 9W6n for acknowledging that our rebuttal is helpful and the topic of this paper is interesting. We will revise the manuscript accordingly based on your comments and further improve the quality of this work.
> > >
> > > ---
> > > Last but not least, we thank Reviewer 9W6n again for the time and effort devoted and the valuable comments drawn during this review.

---

### Author Rebuttal · Authors · 2023-08-10

We sincerely appreciate all reviewers for their insightful comments and help to improve our paper.

---
We are encouraged that the reviewers acknowledging this work:
- "*the paper is well-written and easy to follow* (Reviewer 9W6n, Reviewer Mxk9, Reviewer inkR)";
- "*the overall method is both simple and effective* (Reviewer Mxk9)";
- "*the performance is: a significant enhancement* (Reviewer rM41) / *good* (Reviewer inkR) / *impressively perfect* (Reviewer Mxk9)".

---
We would like to re-emphasize the main contributions of this work:
- In this work, we study how vision foundation models enable networks to comprehend 2D and 3D environments without relying on labelled data.
- We propose a novel Cross-modality Noisy Supervision framework to effectively and synchronously train 2D and 3D networks with severe label noise, including prediction consistency and latent space consistency regularization schemes.
- Experiments conducted on indoor and outdoor datasets show that our method significantly outperforms state-of-the-art methods on 2D and 3D semantic segmentation tasks.

---
We upload a PDF file that contains two supplementary figures:
- *The **improved main figure** to better demonstrate our framework*
- *The **T-SNE visualization of the feature space** with or without latent space regularization*

We address the concerns below and will incorporate all feedback. We will actively participate in the Author-Reviewer discussion session. Please don't hesitate to tell us if there is anything unclear. Thanks!

---
Last but not least, we thank the PCs, ACs, and all the reviewers again for the time and effort devoted to this review.

---

### Decision · Program_Chairs · 2023-09-21

**Decision:**

Accept (poster)

**Comment:**

Reviewers favor the proposed approach, and the majority vote favors the positive side. Authors provide comprehensive feedback on the concerns or questions raised by reviewers during the rebuttal phase. The additional supplement helps to resolve questions satisfactorily. AC read reviews and comments. Primarily, there was a series of discussions between reviewer rM41 and the authors, and even rM41 agreed with the effectiveness of the proposed approach. As a result, AC recommends the paper's acceptance. It is strongly advised to properly apply the requested modification and clarification in the revision.